# Generalism in Nature…The Great Misnomer: Aphids and Wasp Parasitoids as Examples

**DOI:** 10.3390/insects10100314

**Published:** 2019-09-24

**Authors:** Hugh D. Loxdale, Adalbert Balog, Jeffrey A. Harvey

**Affiliations:** 1School of Biosciences, Cardiff University, The Sir Martin Evans Building, Museum Avenue, Cardiff, Wales CF10 3AX, UK; 2Department of Horticulture, Faculty of Technical and Human Science, Sapientia Hungarian University of Transylvania, Sighisoara Str. 1C., 540485 Tirgu-Mures, Romania; adalbert.balog@ms.sapientia.ro; 3Department of Terrestrial Ecology, Netherlands Institute of Ecology, Droevendaalsesteeg 10, 6708 PB Wageningen, The Netherlands; J.Harvey@nioo.knaw.nl

**Keywords:** adaptation, antifeedants, aphids, chemotype, ecology, evolution, phagy, relaxed selection, specialism, generalism

## Abstract

In the present article we discuss why, in our view, the term ‘generalism’ to define the dietary breadth of a species is a misnomer and should be revised by entomologists/ecologists with the more exact title relating to the animal in question’s level of phagy—*mono*-, *oligo*, or *polyphagy*. We discard generalism as a concept because of the indisputable fact that all living organisms fill a unique ecological niche, and that entry and exit from such niches are the acknowledged routes and mechanisms driving ecological divergence and ultimately speciation. The term *specialist* is probably still useful and we support its continuing usage simply because all species and lower levels of evolutionary diverge are indeed specialists to a large degree. Using aphids and parasitoid wasps as examples, we provide evidence from the literature that even some apparently highly polyphagous agricultural aphid pest species and their wasp parasitoids are probably not as polyphagous as formerly assumed. We suggest that the shifting of plant hosts by herbivorous insects like aphids, whilst having positive benefits in reducing competition, and reducing antagonists by moving the target organism into ‘enemy free space’, produces trade-offs in survival, involving *relaxed selection* in the case of the manicured agro-ecosystem.

## 1. Introduction

This year marks the centenary of the death of the great German biologist, evolutionist, proto-ecologist and scientific artist, Ernst Haeckel (1834–1919), who, having read Charles Darwin’s *Origin of Species* [1] formulated the concept of ‘ecology’ in his 1866 writings [2]. Later, following on from the realisation by the English ecologist Charles Sutherland Elton (1900–91) of the existence of the ecological ‘niche’ [3], [defined as the interactions of an organism in its habitat with its surroundings, abiotic and biotic, including other organisms directly or indirectly competing with it], the Anglo-American ecologist, George Evelyn Hutchinson (1903–91), defined it further as ‘*A highly abstract multi-dimensional hyperspace in which the organism’s needs and properties were defined as dimensions.’* [4,5].

In light of these advances in our understanding, it is clear that when a species population A evolves to become species population B, for whatever reason, including chromosomal, then that is clearly the *act of specialisation* and it is a unique event. No other organism fills that niche, except the organism in question and it will remain in that niche until it becomes extinct, is displaced for whatever reason/s, or evolves to fill another niche. Hence, the organism/species population, as it evolves to fill a new niche in time and space, effectively creates a new niche, perhaps for the first time ever, or re-fills a vacant, previously filled niche (e.g., albatrosses vs. pterodactyls, dolphins vs. ichthyosaurs), or displaces an extant and resident organism in an invasive manner.

Furthermore, because of the inherent uniqueness of the ecological niche, an organism can never be truly defined as a ‘generalist’. It may perhaps be polyphagous to a greater or lesser extent (but see below), yet it is surely never generalist; rather, based on ecological and physiological parameters it always remains a specialist. If a species were truly generalist, then its evolution would stop, as the act of evolving to fill a new niche, perhaps due to *intra*- or *inter*specific ‒ or in the case of asexual aphids (Insecta: Hemiptera: Aphididae), phloem sucking plant parasites ‒ *intra*- or *inter*clonal competition, is the main driving force of evolution itself. With *inter*specific pressure, this may also relate indirectly to predator/parasitoid pressure, and the requirement to evolve into a new ‘enemy free space’, or more exactly, niche [6].

Here, following on from our previous deliberations on this topic [7,8], we discuss the reasons why we do not believe that aphids and their wasp parasitoids (Hymenoptera: Ichneumonoidea: Braconidae) are ever generalist. Indeed, we dispute the long-held view that many of them are even polyphagous *sensu stricto*, mainly because of our ignorance in such matters, i.e., lack of empirical DNA evidence. In fact, as far as aphids are concerned, very few species appear to be highly polyphagous (e.g., < 1% of ~ 4000 species recorded worldwide), more especially pests such as the globally important peach-potato aphid, *Myzus persicae* (Sulzer) [9,10,11,12]. In this particular species, the ability seems to be a rather unique specialisation, as most aphid species do not show such ecological-genetic-biochemical flexibility [13,14], or because they actively avoid certain forbs or will die on them if constrained to do so (e.g., *Plantago* spp., see *Discussion* below).

We also suggest that the *apparent* extensive polyphagy seen by many pest aphid species on crops is the direct result of *relaxed selection* arising from a reduction in secondary plant antifeedant titre in artificially designed and reared crops within the manicured agroecosystem, cossetted plants that would normally be unsuitable as aphid hosts in their wild form. In a truly wild scenario, both chemical defences and predator pressure would exclude such flexible behaviour and would limit the insect to their original, long co-evolved natal host/s, either a single host or taxonomically closely related hosts within the same plant family.

## 2. Background

The term generalist and specialist are still widely used in the scientific literature, usually in reference to diet breadth (e.g., [15,16,17,18,19,20,21,22,23]). However, there is usually little empirical support for the contention of generalism, especially from the point of view of the many eco-physiological factors that are crucial in realistically evaluating an animal’s true food preferences within its natural environment, whether we are referring to herbivores or their natural enemies, such as predators or parasites [24,25]. Furthermore, this also applies to the micro-habitat of a single plant or a larger, more complex and heterogeneous environment, like deciduous woodland [26]. Finally, there are no molecular ecological data (nucleotide or enzyme) that broadly support generalism as a concept [7], let alone a reality, perhaps with the exception of recent sequencing studies by Mathers et al. [13] of a few asexual lineages of *M. persicae*.

In our previous paper [7], we emphasised the improbability of generalism in nature, with special reference to insects, including the possibility of the existence of arrays of morphologically-similar/ identical cryptic species only identifiable using high resolution molecular markers, especially DNA markers. In a later paper [8], we extended the discussion in relation to diet breadth, arguing that if generalism really does exist, it may be defined as four broad categories. In the present paper, we go further and suggest that the entire notion of generalism should be abandoned from its present-day widespread usage, more especially because of the lack of empirical support, and be replaced only with notions of phagy, *mono*-, *oligo*- and *polyphagy*, preferably reinforced with real molecular and ecological data.

This is not merely a semantic argument. This is because to speak of generalism is to misunderstand the precise definitions of ‘niche’ and the specific multivariate ecological factors—abiotic and biotic—that an organism faces and satisfies in order to fill such a niche. These factors may change in time and space to varying degrees. For example, when a herbivorous organism invades a new range and encounters new plant hosts, these may either not be adapted in terms of defence, be that physical (spikes, sticky trichomes, etc.) or chemical (i.e., secondary plant antifeedants), or involve indirect defence agents such as insect predators and parasitoids that are attracted to herbivore-induced-plant-volatiles, e.g., [27]. These agents may initially be highly effective in their effects until some kind of ‘ecological equilibrium’ is attained or the organisms in question cannot survive in the new environment (for whatever ecological reasons) and dies out, either rapidly or slowly and in part or all of its newly invaded range.

The term generalism is not a substitute for polyphagy, nor in some sense is specialism for lower levels of polyphagy. This is because of the fact that the broad terms generalism and specialism can be subdivided, as we discussed earlier [8]. In other words, these terms only have validity when thus subdivided and even then, they somehow evade the truth that species fill specialised ecological niches, which is what species populations in both categories do. In predatory insects, perceived generalists like the invasive Asian Harlequin ladybird beetle, *Harmonia axyridis* (Pallas) (Coleoptera: Coccinellidae), which appears to be a highly opportunistic and polyphagous predator, even eating other ladybird species [28], is nevertheless constrained it terms of its diet and hence diet breadth by the unique features of its ecology, anatomy, gut enzymes, etc. For example, it can only tackle arthropod prey within the confines of its ability to deal with them physically, whilst it tries to seize and devour them, and later digests them, a process ultimately leading to co-evolution [29] of a limited preferred range of prey (see also below, *Evolutionary trends*).

Of course, it is not possible to survey the whole of the animal kingdom to support our views on this matter, but in the case of aphids and wasp parasitoids, our own fields of speciality, we provide examples of why so-called highly polyphagous species are in actuality much less so. Furthermore, with aphids at least, this specialism is, as we have stated before [8,30], usually confined to host plants within the same taxonomic family. In this case, we propose that enzymic/metabolic systems [31] and recently discovered epigenetic mechanisms [32] that are used to encourage or deter/inhibit feeding are probably common/similar across host plant species (since they have a common ancestor by descent). Consequently, in an adaptive sense, host transference of aphids to a closely related host plant species is probably not then so much of a genetic-molecular-biochemical challenge (cf. also [14] and references therein and [33] in the case of the whitefly, *Bemisia tabaci* (Gennadius) species complex; Hemiptera: Aleyrodidae)). Advantages of such host plant transference, especially in the natural or ‘wild’ environment’, allows the animal in question to escape into ‘enemy free space’ [34,35], allopatrically or sympatrically, even though there are undoubtedly fitness costs involved in terms of impaired reproductive fecundity and hence population growth rate (see below). It also allows the animal in question to escape competitive forces from other herbivores, predators, etc. attacking and exploiting the same target organism/s, and hence same resource, e.g., [36].

However, eventually, as aforementioned, given enough time, antagonists (predators, prey and pathogens), may adapt to the herbivore on its new plant host/s and this could prove fatal to it or prevent its expansion. Yet in the agroecosystem, such selective/ competitive pressure may be significantly reduced in comparison with the natural habitat/ecosystem that the pest originally evolved in. In this sense, it is not subject to the full panoply of negative selective forces that would normal impinge on it, despite having the benefit of a large mass of nutritious food available, usually more so than the original host/s on which it evolved in pre-agricultural times. On the other hand, it is also negatively impacted directly by ‘generalist’ or ‘specialist’ pesticides used against it, often the latter in recent times (e.g., the carbamate Aphox^®^ (Syngenta; Cambridge, UK) for aphids), and natural biological control agents applied or enhanced within the agro-ecosystem, usually invertebrate predators and parasitoids, although viral/bacterial and fungal pathogens may be utilised within the outdoor agro-environment, but more normally within the confines of the glasshouse [37,38].

Thus the agro-ecosysem has its advantages for pests, including those invading new regions and crops, but ultimately, they are also subject to negative selective forces [14,36]. If so, the balance is perhaps only in favour of those pests that show high mutation rates as well as high reproductive rates in order to keep pace ahead in the ongoing arms race that is modern farming-horticultural-forestry practice, including use of one or more insecticides, to which the more polyphagous species of aphid may in terms of detoxification be to some large extent pre-adapted [14,39]. Meanwhile, pests are also threatened with unsuitable/deterrent strains of crop plants. This is either achieved through conventional breeding and rearing of crops to display enhanced plant resistance qualities [31,36] or these days, genetically modified (GM) crops that express higher titres of ‘bottom up’ secondary plant antifeedants [40], kairomones that enhance ‘top down’ antagonist numbers thereby reducing the impact of pests [41,42,43], or the expression of toxins such as Bt (*Bacillus thuringiensis*) [44].

## 3. Evolutionary Trends: The Thrust of Ecological Diversification

The entire thrust of evolution concerns adaptive radiation leading to ecological specialisation and ultimately to full speciation, perhaps via intermediate stages of lower levels of divergence, i.e., as subspecies, sibling species, sister species, races, biotypes, ethotypes, chemotypes, etc., [45,46]. Such ecological specialisation is often manifested as either allopatric, parapatric or sympatric divergence, due to one or more genetic (minor and major, i.e., karyological), chemical/biochemical, morphological and behavioural modifications of the animal in question [47,48]. Either way, the net result generates finer and finer-grained diversification of the original natal species population to fill available ecological niches, and perhaps to create novel ones. This kind of radiation is now well documented following research of an array of living organisms, more especially insects, e.g., aphids [30] and wasp parasitoids [7,8], and indeed vertebrates like amphibians [49] and reptiles [50]. This trend, one seen throughout the course of the history of life on Earth going back to the most remote geological periods and exemplified by such ancient diverse taxa as trilobites [51], dinosaurs [52] and more recently, birds and mammals [53], is ubiquitous and its predominance as the main driver in the evolutionary process seems beyond question.

## 4. Aphids

Here, by way of an example of apparent polyphagy, we use the published data set provided by Tatchell et al. [9], in which 30 insect pests representing species regularly captured and recorded in the UK nationwide Rothamsted Insect Survey using 12.2 m high suction traps are described [54]. We concentrate only on the ten most polyphagous of these in terms of the extent of dietary breadth (those species that are not polyphagous on plants within the same taxonomic family, but rather between plant families (Table 1)). One species, *Myzus ascalonicus* Doncaster, appears to be obligatorily asexual with no sexual forms (males and egg laying females = oviparae) ever having been found anywhere in the world, whilst *M. ornatus* Laing also seems to be largely parthenogenetic worldwide, perhaps with the exception of populations in the Himalayas [11,55].

The remaining eight species we discuss are either heteroecious (host alternating) and complete an annual sexual phase (holocycly) to varying degrees, or even occasionally produce sexual forms on their secondary herbaceous hosts [55]. Some of these species have a preferred woody host to which the winged males and pre-sexual winged females (gynoparae) return in the autumn under the influence of short day length and low ambient temperature conditions to mate and lay overwintering eggs [56]. The plant hosts are principally spindle, *Euonymus europaeus* (Celastrales: Celastraceae) in the case of the black bean aphid, *Aphis fabae* Scopoli, and several members of the Rosaceae, principally *Prunus* spp., in the case of *Brachycaudus helichrysi* (Kaltenbach), *Hyalopterus pruni* (Geoffroy) and *M. persicae* (notably also peach, *Prunus persica* in the last species), and *Malus* spp. in *Dysaphis plantaginea* (Passerini) [11,55,57].

According to Blackman [55], *Myzus certus* (Walker) produces some sexual forms on Violaceae (*Viola tricolor* L.), although this aphid species appears to be predominantly asexual in the UK, as are *M. ornatus*, *M. persicae* and *Aulacorthum solani* (Kaltenbach) due to the mild winter climate generally prevailing there [58]. Both *B. helichrysi* and especially *M. persicae* are much more restricted in terms of host associations on their primary woody hosts than on their secondary herbaceous spring and summer hosts [7,9]. *H. pruni* clearly has morphologically similar host-adapted strains/sub-species on *Prunus*, including apricot, blackthorn, plum and cherry, respectively [59]. Blackman [55] uses the term polyphagous in describing aphids with an apparent wide host range (cf. also [9] and [11] for further details of host associations of the aforementioned species and [60] with regard to their molecular phylogeny within the tribe Aphidini that includes them).

From this table, it is clear that even in these apparently polyphagous species, their diet breadth is not as large as hitherto assumed, being on average around 4 plant species per plant family. These findings suggest that these species are seemingly constrained to a relatively few host plants, most likely in the same or closely related families, so that the ecological challenge, especially that related to chemical antifeedants, and perhaps also antagonists like hymenopterous parasitic wasps, is thereby reduced.

In order to highlight such constraints in aphids, which have been more intensively studied than many other herbivorous pest insect species to date, it is instructive to look at grass and cereal-feeding aphids of the genus *Sitobion*. In the UK, two species are found commonly on such hosts. These are the blackberry-grain aphid*, Sitobion fragariae* (Walker), a holocyclic species which host alternates between its Gramineous summer host/s and a primary woody host, especially blackberry, *Rubus fruticosus* L. agg., and the grain aphid, *S. avenae* (F.), a predominantly anholocyclic (asexual) pest species of cereals, which remains all year on its gramineous host/s, although can occasionally produce sexual forms and lay cold hardy overwintering eggs in cold winters [11,61]. Using allozyme markers (peptidase-1), Loxdale & Brookes [62] demonstrated that > 90% of the aphids on grasses (cocksfoot grass, *Dactylis glomerata* L.) were *S. fragariae*, the remainder *S. avenae*. Hence, *S. avenae* is seemingly not that well adapted to this wild grass host as *S. fragariae*, but will go onto it. There is also evidence from DNA markers (microsatellites and mitochondrial DNA) of introgression between the two species [63]).

The hydroxamic acid DIMBOA (2,4-Dihydroxy-7-methoxy-1,4-benzoxazin-3-one), is involved in aphid resistance of cereals and is known to have such an effect via the inactivation of α-chymotrypsin, esterases, glutathione S-transferases, UDP-glucose transferases (involved in the detoxification of phenolic aglucones), and acetylcholineesterases [64,65,66]. Other enzymes are also known to be involved in plant resistance, i.e., Phenylalanine ammonia-lyase (PAL), polyphenol oxidase (PPO), and peroxidase (POD), all enhanced upon cereal aphid attack. [67]. In addition, DIMBOA is known to affect several aphid organelle marker enzymes. Thus catalase from peroxisomes and cytochrome c oxidase from mitochondria increased their activities around two fold. These enzymes are involved in metabolizing xenobiotics by aphids [65]. Lastly, using molecular approaches, two dominant aphid resistance genes have been isolated to date. Both encode nucleotide-binding site leucine-rich repeat (NBS-LRR) proteins involved in the specific recognition of aphids by cereals. [68]. As stated by the authors ‘…*most aphid resistances were shown [to be] biotype specific*.’ Hence it is clear that these agents collectively act in both deterring aphids by inactivating key enzymes involved in detoxifying plant antifeedants, whilst they also boost the aphid’s own enzyme involved in tackling xenobiotics, so that ultimately they tend to constrain aphids to long co-evolved relationships with the plants they attack, in this case members of the Graminaceae (Poaceae). van Emden [31] discusses the basis of host plant resistance in aphids*;* cf. also [69].

That such relationships have validity in the agroecosystem is demonstrated by the fact that cereal aphids such as *Sitobion* spp. have clear host preferences as shown using high resolution molecular markers (RAPDs, random amplified polymorphic DNA) (e.g., [70]), even to the extent that winged asexual female migrants (here *S. avenae*) landing on crops in the spring show definite preferences when given a choice of different grasses and cereals [71], whilst it has been known for about 40 years that different cultivars of wheat (*Tritucum aestivum* L.) are differentially susceptible to aphid attack, e.g., cv. Kador, a low yielding ‘blue’ wheat is highly resistant, whereas cv. Maris Huntsman, a high yielding variety, is very susceptible, as can readily be seen on visual inspection. Controlled greenhouse trials using a range of different wheat cultivars confirm this trend ([72]; cf. Table 1 in this paper). Such differences presumably mainly relate to the expression of genes involved in resistance to aphids, including those for DIMBOA titre and aphid recognition of suitable hosts, and other proteins/enzymes involved in plant resistance.

That fact that aphids are most often constrained chemically to certain plant species or even parts of these plants is demonstrated in the case of the aphids attacking the highly chemically defended (with terpenes and terpenoids) plant tansy, *Tanacetum vulgaris* L. About 13 species of aphid are known to attack this plant [55], but only three commonly, namely *Macrosiphoniella tanacetaria* (Kaltenbach), *Metopeurum fuscoviride* Stroyan and *Uroleucon tanaceti* L. [55,73,74]. The first species tends to infest the crown of the plant, including the flowers and new flush leaves, the second species, which is ant attended the stems, and the last species, the underside of the lower leaves [74,75,76]. Benedek et al. [77] were the first to show that tansy aphids, *M. fuscoviride*, were differentially attracted to, and survived on, different genets of tansy, dependent on terpene/terpenoid chemotype (cf. also [78]). In fact, and fascinatingly, recent research by Jacobs & Mueller [75,79,80] have revealed that this fine-grained micro-ecological distribution of tansy aphids between plants also even extends within plants, where it is again governed by the *intra*-plant chemical profile, as well as trichrome arrays (see *Discussion* for further details).

These examples of oligophagous aphids, here cereal aphids confined to hosts within the same plant family, Graminaceae, and monophagous tansy aphids, holocyclic on tansy [55], demonstrates the close, indeed highly chemically-constrained nature of the specialism of these insects, a relationship which is seen to be increasingly fine-grained, and indeed was unsuspected until the recent application of molecular and chemical analytical techniques.

## 5. Parasitoid Wasps (Hymenoptera)

Aphid parasitoids are among the best studied in the parasitic Hymenoptera, presumably because many aphids are serious pests of crop and ornamental plants grown both in the field and the greenhouse [81]. The vast majority of aphid parasitoids are endoparasitic koinobionts in the Aphidiinae [82] and thus attack different stages of aphids that continue feeding and growing during the course of parasitism [83]. In holometabolous insect herbivores that complete four stages of metamorphosis, the feeding stage (larvae) very often possess potent immune defences [84,85]. Endoparasitoids of these herbivores have evolved a suite of intricate strategies to evade or abrogate these immune systems including the use of potent venoms, or the expression of symbiotic viruses and virus-like particles [86,87,88].

In contrast, in hemimetabolous insects, where the young nymphs resemble the adults but lack wings, metabolic defences are often rudimentary. However, many aphid species have evolved mutualistic relationships with endosymbionts that confer resistance against pathogens and endoparasitoids [89,90]. The presence of metabolic defences in both holo- and hemimetabolous insects, irrespective of cause, has led to strong co-evolutionary relationships with endoparasitoids that in turn promotes and reinforces their specialization. For instance, the host range of many endoparasitoids of lepidopterous hosts is limited to a single family or even species [91]. Moreover, because every species of host has its own well-defined niche, based on for example, habitat preference and dietary breadth especially, parasitoid wasps have become highly specialized in exploiting these species-specific niches. Aphid parasitoids (Braconidae: Aphidiinae) are no exception.

In the most comprehensive study of aphid primary and secondary parasitoid communities yet undertaken, Müller et al. [92] analysed species found in an abandoned grassland in southern England. They compared primary and secondary hyperparasitoids and mummy parasitoids associated with 23 aphid species that were collected over two years in a single damp field site. Most of the parasitoids only attacked a small number (i.e., 1–5) of aphids, with a single parasitoid species, *Praon dorsale* (Haliday), being recovered from eight aphid species, i.e., just over a third of the number of species sampled. The host range of hyperparasitoids and mummy parasitoids was broader, but by no means did they attack all species of primary parasitoid. Bear in mind that the study was performed in a small, localized habitat, such that most of the species were spatially if not temporally sympatric. The study reveals that a range of factors have driven varying degrees of specialization among the parasitoid/hyperparasitoid/mummy parasitoid community.

Some aphid endoparasitoids, such as *Aphidius colemani* Viereck are widely used as biological control agents against a wide range of aphid species [93]. However, even this apparently generalist species tends to prefer some aphid species over others [94,95]. Moreover, direct rearing data, as in the case of the parasitoid *Diaeretiella rapae* (M’Intosh), feeding on a range of aphid hosts collected from wild and cultivated plant hosts [96], also provides clear evidence of both ecological specialisation and the existence of cryptic species. An important point to stress is that aphid parasitoids do not attack insects in any other taxonomic groups. So any species in the genera *Aphidius*, *Praon*, *Diaeretiella*, *Ephedrus* or *Aphelinus* that attacks more than a single host species will still be restricted to aphids. We emphasize strongly that terms like ‘generalism’ and ‘specialism’ thus need to take into consideration the scale to which they are applied. Aphid parasitoids are all highly specialized on a subset of aphid species.

## 6. Discussion

The whole debate about generalism *versus* specialism is highly pertinent to discussions concerning the cutting edge of the ecological-evolutionary processes. This is because it is the act of host switching to a new food resource, often in the face of competition of one form or another [97], or perhaps under predator/parasitoid pressure to escape into enemy free space [6], that is a crucial element in the act of diversification and adaptation.

As we relate in the case of aphids and their braconid wasp parasitoids, both aphids and parasitoids have their clear habitat preferences and diet breadth. In aphids these aspects are governed, indeed constrained, by plant chemistry especially and plant anatomy evolved over long periods of time, perhaps many millions of years. Because of these constraints, it has led to inter-actions and interrelationships such that most species (> 99%) are indeed specialist [10,30]. This specialism is also to some degree governed by secondary (facultative) endosymbiotic bacteria [98,99,100,101]. That the co-evolution of such specialism is indeed ancient and developed prior to human agricultural practices is also evident in some cereal aphid species studied (see below). Lastly, that the main pest aphid species in the UK are agricultural pests provides support to the idea that relaxed selection in the agroecosystem due to selective breeding of high yielding crops with reduced titres of unpleasant tasting antifeedants, including to humans and their livestock and indeed phytophagous insects such as aphids (as seen in wheat *cv.* Maris Huntsman in the case of cereals) has allowed these animals to exploit this relatively poorly chemically-protected resource and thrive accordingly [14]. Indeed, some aphids have thereby been able to extend their host range, thus diet breadth, and perhaps greatly so as in *M. persicae* [13].

With wasp parasitoids, the degree of specialism and hence diet breadth is constrained by immunological competence of the aphid under attack and its a ability to encapsulate its antagonist’s egg/young larva, in turn dependent on the species of secondary endosymbiont involved which confines the actions of the antagonist and selects for the co-evolution of precise aphid species-parasitoid genotype relationships [102]. By such means, the parasitoids, as with the aphids, are not ‘free agents’, a fact that restrains their abilities to host shift and ultimately to broaden their host breadth. In essence, this means in relation to the two groups of insects we have chosen to discuss, that because they are constrained in terms of diet breadth and may indeed have a smaller one than imagined, then *ipso facto* they cannot be generalists. This we believe is strong support for our notion that the term should be dropped in favour of levels of phagy. Furthermore, since we don’t know the true level of cryptic speciation in both various aphid species and their wasp parasitoids *sensu lato*, and since this can only be determined in follow-up studies using molecular markers, it is better to veer on the side of caution and consider it more plausible for a given species to have a restricted diet breadth rather than an extended one.

That selection of asexual lineages (clones *sensu lato*; [103]) occurs in the field has been demonstrated using polymorphic microsatellite markers [104], some such selection related to host in the case of *S., avenae* [105]. Besides direct selection of aphid genotypes due perhaps to direct *intra*- and *inter*clonal competitive interactions, antagonists can also shape the nature of the distribution of aphids on plants, thereby effectively promoting host switching into enemy fee space. For example, in the pea aphid, *Acyrthosiphon pisum* Harris, the host-adapted red and green forms of the aphid appear to be under differential host related predator and hymenopterous parasitoid pressure, this thereby governing the ability of these colour forms to switch hosts within the agroecosystem [6]. Furthermore, insecticide applications select for particular resistant aphid genotypes in time and space and by so doing, structure aphid populations [106,107,108]; reviewed in [109]. Lastly, over both short-term (i.e., human agricultural times scales, ~ 8000 years B.P.; [110]) and much longer geological timescales, plant resistance genes have played a decisive role in shaping the host associations of aphids, often limiting them to certain varieties of crops and thus preventing them from attacking other related species or cultivars. The latter are overcome by the evolution of anti-plant resistant mechanisms in aphids, for example, biotypes of the greenbug aphid, *Schizaphis graminum* (Rondani) overcoming the resistance mechanisms of resistant strains of sorghum, *Sorghum bicolor* ([111,112,113]; cf. also [36] in the case of the soybean aphid, *Aphis glycines* Matsumura).

Returning to the broad concept of generalism, more especially in the manicured semi-real world of the agro-ecosysem, it is seen from Table 1 that when the total number of plants infested by one of the ten species of pest aphid (S) is divided by the number of plant families (F), the resulting ratio is actually quite small, around 4.0 *plant species per plant family*. As this includes seven highly polyphagous species (*A. fabae, A. solani, B. helichrysi, M. euphorbiae, M. ascalonicus, M. ornatus and M. persicae*), these findings suggest that these apparently highly polymorphic species are in reality not as polymorphic as first appears. If so, this implies that there are indeed strong selective constraints on host adaptation in aphids, probably principally secondary plant antifeedants and antagonists, but also doubtless related to host plant odours, which attract the pest to the crop, as recently shown in Swede midge (*Contarinia nasturtii* (Kieffer); Diptera: Cecidomyiidae), a specialist pest of brassicas. Here, the attractiveness of non-host odours modifies the insect’s behaviour, and is negatively correlated with phylogenetic relatedness, thereby reinforcing its specialism (cf. [114] for further details).

Some mention should also be made of phenotypic plasticity in relation to host shifts enabling expansion of diet breadth in insects [115]. It may be that this always has a genetic basis [7]. Certain pest aphids appear to show phenotypic plasticity, allowing them to attack a range of host plants [14,116,117,118,119]. Even so, host switching to a non-natal host has associated fitness costs in terms of development rate and fecundity, as has been shown in several studies [120,121] (but cf. [122] in the case of laboratory studies on the North America blue butterfly, *Lycaeides (Plebejus) melissa* W.H. Edwards; Lycaenidae: Lepidoptera). From the sparse empirical data available, as aforementioned, only certain asexual lineages of *M. persicae* tested experimentally appear to be polyphagous in that their development rate and fecundity is apparently not impaired on transference to other host plants [13]. But this seems to be very much the exception rather than the rule [30]. Moreover, there are plants that appear to be resistant to *M. persicae*, including ribwort plantain, *Plantago lanceolata* L. and warty cabbage, *Bunias orientalis* L. (J.A.H., pers. obs). The latter is a close relative of plants that are highly susceptible to this aphid, such as black mustard, *Brassica nigra* L. Probably the duel effects of plant antifeedant and different amino balance of the plant phloem affect the fundamental biochemistry of the aphid via its secondary endosymbiont/s: these co-evolve with their aphid host and are crucial in terms of its host associations (cf. [14,30] and references therein).

In addition, there are other negative forces at work with regard to switching host, as already discussed, some positive (enemy free space), some negative (competition with existing herbivores in the case of aphids, possibly other aphid species or even genotypes of given aphid species, e.g., [104]). Competition probably explains why on tansy, although the plant can support up to 13 aphid species, three commonly, these display different distributions on the plant and tend not to co-occur, primarily feeding on the flower stalks of the plant, stem, and underside of lower leaves, respectively [74,75]. This lack of co-occurrence among the three aphid species may be due to the fact that only *M. fuscoviride* is ant-attended, especially by *Lasius niger* (L.), *Formica rufa* L. and *Myrmica rubra* (L.). These ants appear to prevent other aphid species from settling on plants occupied by *M. fuscoviride,* perhaps even killing them [30,123], although not necessarily consuming them because of the terpenes/terpenoids [124] that they contain (*cf*. [77,78] and references therein). In addition, the distribution of trichomes and sugars, organic acids and amino acids, and secondary plant antifeedants, especially terpenes/terpenoids, has recently been found to significantly influence the positioning of different tansy aphid species on and between plant chemotypes [75,77,78,79], and as such and as Jakobs & Mūller state in their 2018 paper in relation to tansy aphid feeding preference and performance [75]: *“These different performance optima may cause niche differentiation and, therefore, enable co-existence. In conclusion, the tremendous variation in plant chemistry even within one species can affect the distribution of highly specialized aphids at various scales aphid species-specifically*.” (cf. also [80]).

It is probable that these effects of plant chemistry on tansy aphid *intra-* and *inter*plant distribution, including molecular ecology as reflected in specific microsatellite MLG distribution, occur at the recognition and settling phase of the aphids during their inter-plant migration in July [78,125,126]. This appears to be in direct relation to visual and plant volatile cues prompting settling and probing-feeding [127]—as earlier shown with the grain aphid. *S. avenae* by Lushai et al. [71]—as well as the presence of ants preferentially associated with certain plant chemotypes in the case of tansy-feeding aphids [78].

## 7. Conclusions

In summary, we believe that for the reasons outlined above, the term ‘generalism’ is a misnomer and should be discontinued by aphidologists and other entomologists/ecologists in favour of levels of phagy in descriptions of the dietary breadth of aphids and wasp parasitoids and indeed other insects, be they herbivorous, carnivorous, detritivores or parasitic. The term specialist is more acceptable because this is what animals and indeed all living organisms to a large extent are, filling their own unique ecological niches, and often being highly constrained in terms of lifestyle and diet breadth due to a plethora of governing morphological-genetic-chemical-biochemical-physiological and behavioural factors (essentially the abiotic and biotic vectors, governing factors in the Hutchinsonian concept of a given organism’s niche in time and space). When a herbivorous or carnivorous species expands its dietary breadth, then populations become isolated and specialized on new hosts/food, reducing or even eliminating gene flow among populations feeding on different hosts. So an increase in dietary breadth creates specialism at both the population and eventually species level. However, because there are levels of specialism ‒ for example, an insect attacking a single plant or closely related plants in the same family or only the fruit ‒ then describing diet based on differing levels of phagy is a more realistic approach, or so we feel. Indeed, ecologists are always using terms to define categories of traits, such as ‘habitat or niche specialist’. Hence there is no reason why the use of similar terminology defining levels of phagy cannot be universally applied to insects, or indeed any animal taxa. 

The fact that recent studies, often using chemical and or molecular biological markers, provide further evidence for finer and finer gradation of ecological specialisation, especially in well-studied groups such as aphids (e.g., [78,79], and other animals like amphibians, supports the notion of increasing specialisation, not the reverse. Lastly, the fact that modern taxonomic/systematic assessments of various taxa tend to *split* species into lower levels for evolutionary divergence is proof of this. Even families and genera themselves are continually being re-appraised, sometimes causing familiar groupings to be re-classified following the acquisition of greater understanding of their morphology, genetics, molecular genetics, biochemistry/chemistry and of course ecology and behaviour, e.g., [128].

## Figures and Tables

**Table 1 insects-10-00314-t001:** Extent of polyphagy in ten major aphid pests of UK crops in terms of the number of plant families (F), and total number of plant species (S) they have been recorded on, including overwintering woody hosts, and the ratio of S/F.

Aphid Species	No. Plant Families Recorded (F)	Total No. of Plant Species Recorded (S)	Ratio S/F
*Aphis fabae* Scopoli *	71	293	4.12
*Aulacorthum solani* (Kaltenbach) †	59	196	3.30
*Brachycaudus helichrysi* (Kaltenbach) ***	20	119	5.95
*Dysaphis plantaginea* (Passerini) ***	3	10	3.33
*Hyalopterus pruni* (Geoffroy) ***	8	21	2.63
*Macrosiphum euphorbiae* (Thomas, C.A.) †	63	265	4.21
*Myzus ascalonicus* Doncaster ††	36	129	3.58
*Myzus certus* (Walker) †	4	18	4.50
*Myzus ornatus* Laing †	41	136	3.32
*Myzus persicae* (Sulzer) †	72	305	4.24
**Mean ± SE of n = 10**	**37.7 ± 8.8**	**149.2 ± 35.8**	**3.9 ± 0.29**

Key: * Heteroecious (host alternating) and holocyclic (i.e., with one annual asexual phase in-between many asexual spring and summer generations), with one or more woody overwintering host; † predominantly asexual (anholocyclic) in the UK; †† Obligate asexual globally (as for as is known).

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
