# Peer review of "Generalism in Nature…The Great Misnomer: Aphids and Wasp Parasitoids as Examples"

_insects, 2019, doi:10.3390/insects10100314_

Round 1
Reviewer 1 Report
The article Generalism in nature...the great misnomer: aphids and wasp parasitoids as examples, provides sufficient background and includes relevant references. Besides the conclusions are supported by the results, and it presents a high Originality / Novelty and an Overall Merit.
My unique suggestion is below:
Lines 82 – 83: The authors make mention to molecular ecological data (DNA or allozymes) that could support generalism as a concept. Could you cite in details which techniques could support it?
Author Response
Referee 1
We thank the referee for reading our manuscript and giving us the benefit of their wisdom in terms of its suitability of publication. We are naturally pleased that the referee is in agreement that this is an important topic for discussion and hence publication in the Cutting edge evolution of insects special issue of Insects and hence in the public forum
“…[the article]…provides sufficient background and includes relevant references. Besides the conclusions are supported by the results, and it presents a high Originality / Novelty and an Overall Merit.”
We thank the referee for pointing out the issues with Lines 82 – 83: ‘The authors make mention to molecular ecological data (DNA or allozymes) that could support generalism as a concept. Could you cite in details which techniques could support it?’
In answer to this, we cite the only paper that we know relating to this issue in aphids, namely by Mathers et al. [ref. 13, loc. cit.]
We have changed this accordingly to:
“Finally, there are no molecular ecological data (nucleotide or enzyme) that broadly support generalism as a concept [7], let alone a reality, perhaps with the exception of recent sequencing studies by Mathers et al. [13] of a few asexual lineages of M. persicae.”

Reviewer 2 Report
In their manuscript Loxdale et al. argue against the use of the term ‘generalist’, or at least as a commonly accepted substitute for wide dietary breadth or polyphagy. In this regard this manuscript follows the pathway of the previous publications from the main author but goes further by providing examples to support the aforementioned claim. Although the authors attempted to connect them logically, in my opinion, the manuscript discusses two distinctive issues; that the real diet breadth of some taxa is narrower than we previously thought, and the argument for abandoning the use of the term ‘generalist’ or ‘generalism’. Although the main topic of the manuscript is the ecological background of the ‘generalis’ – ‘specialist’ terminology, in my opinion, the authors did not succeed to build a clear argument around this concept. Rather, the manuscript seems to me as a collection of semi-connected facts, often with hints only how they are related. Whereas in their previous works they made a better job to support their arguments, in this manuscript I struggled to find a clear story. Neither the somehow unusual ‘hybrid’ structure of the manuscript, nor the often long, nested sentences provide a clear and easy way for the reader to understand the authors’ point of view. Particularly the Discussion seems to be no more than listing a few more (strictly!) aphid examples without a full synthesis, or crystallizing the main message. Moreover, since the journal’s target audience is not theoretical ecologists or evolutionary ecologists, I believe, that several points of the argument need further explanation and/or more background.
Nevertheless, the manuscript discusses an interesting topic. Although my personal opinion is not fully in line with that of the authors (please see some points below), I still think that publishing the manuscript can trigger a healthy scientific discussion.
I suggest the authors to revise the text from the logical point of view, consider a structural change (maybe abandon the current Introduction-Background-Evolution-Aphids-Parasitoids-Discussion structure), break up long sentences and delete irrelevant content (please see some suggestions in the text). As a minor note, I also would suggest the authors to standardize what typographical element they use to highlight the terms ‘generalist’ and ‘specialist’ (e.g. italic or quotation marks).
My comments on the topic (this should not influence the Editor’s decision):
In most of the cases ‘generalism’ refers to more than simply polyphagy or diet breadth. It, indeed, refers to a set of life-trait characteristics (e.g. habitat preferences) in which a species shows a wide level of toleration, or just a general broad niche breadth, or euryec species. When the authors talk about the evolution pointing to specialization they ignore the fact that from this perspective none of the life-traits are constant, even the newly evolved ‘specialist’ species will (or may be) a different one than the previously ‘generalist’. In evolutionary terms species are not constant entities and, thus, neither are any properties linked to them, what we use are snapshots of time. Therefore, arguing for avoiding the use of one term because it changes over an evolutionary time scale is like arguing against using the expression ‘species’ (which, by the way, also would be a valid argument). I believe that the ‘generalist’ – ‘specialist’ grouping has an undeniable practical side for ecologists working on extant organisms (i.e. on a short temporal window on an evolutionary scale). In many cases diet breadth is unknown even to that level that would allow us to decide if a species is oligophagous, or polyphagous, we only know it is not monophagous. In these cases the above grouping has its own merit. Although subdividing these categories is still informative, subdivision of any categories can be done infinite times; thus the question is whether these subdivisions provide significantly more insight into a problem or not. Biotic or abiotic constrains, often locally only, limiting a population’s (!) realized feeding niche cannot be a valid argument against species’ polyphagy. Phagy, or any other life-traits, are linked to species, not to populations, and thus the entire host range should be considered from all populations. Host choice for any type of organisms is a balancing act between availability and suitability. This is true even for species with very much specialized diet only that in this case the choice is not between host species but between host individuals. Therefore, fitness costs associated to host-plant switches do not support host specialization, and thus the need to re-term ‘generalist’.

Author Response
Referee 2
As with Referee 1 & 2 , we are grateful to the referee for reading our manuscript and giving us the benefit of their wisdom in terms of its suitability of publication. We are naturally pleased that the referee is in agreement that this is an important topic for discussion and hence publication in the Cutting edge evolution of insects special issue of Insects and hence in the public forum
‘Nevertheless, the manuscript discusses an interesting topic. Although my personal opinion is not fully in line with that of the authors …….I still think that publishing the manuscript can trigger a healthy scientific discussion.’
We agree with the reviewer that the MS was somehow not as clear as our previous two papers published in the Biological Journal of the Linnean Society in 2011 and 2016, respectively, and that it could be improved in terms of both its message and general clarity. We have therefore endeavoured to extensively prune and revise the manuscript, including deleting certain sections (e.g. on mammals) and concentrating predominantly on insects, more specifically our two chosen groups, Aphids and Wasp Parasitoids, and have added some new sections and new references along with deleting some previously cited references. Even so, we with have tried hard not to go ‘off-piste’ too radically, i.e. away from our original broad vision of what this article is about. Hence our decision to retain the original headings and broad structure. Thanks to the Referee’s wise comments, we have seriously reconsidered the text and its message and amended it accordingly. We believe that we have achieved this and naturally hope and trust that the referee agrees and that the paper is now acceptable for publication.
The topic is complex and multivariate (in a Hutchinsonian sense) and we are only talking primarily about aphids and wasp parasitoids and in this context, we think that diet breadth and concepts of generalism are integrally connected and intertwined. If seems from our simple re-analysis of Tatchell et al’s (1983) suction trap data set that even the most ‘generalist’ aphids are not as polyphagous as hitherto thought, attacking only a maximum of ~ 4 plant species per plant family. This to us casts grave doubts on concepts of generalism, which anyway, we believe we have argued convincingly against by showing that in both aphids and wasp parasitoids, all sorts of constraints (chemical-biochemical, physiological, immunological, etc.) make such a scenario unlikely. If an aphid or parasitoid does try to expand its diet breadth, even if this potentially has advantages for it (e.g. entering ‘enemy free space’), then because of the long co-evolution between plant & plant parasite and aphid & wasp parasite, respectively, then this is difficult to achieve.
Plants, the greatest chemists in the world, have, over unimaginable aeons, produced a plethora of chemically complex antifeedants, honed largely to deter specific herbivores, including aphids. It is difficult to believe that aphids can tackle (i.e. degrade, neutralise, etc.) all such chemicals; rather they specialise on single plant hosts with a restricted ’armoury’ of antifeedants or similar ones within closely relate plant families, where the chemical-ecological challenge (including combatting antagonists) may be less.
We humans grow crops that we like, both in terms of high yield and often with reduced antifeedant titre, because these are more tasty/less toxic to us. But maybe this is true for the aphids too, and hence they can expand their host range more easily in the light of such reduced selective pressure. They may also encounter less co-evolved antagonists as a result. With aphid parasitoids, they are especially constrained in terms of diet breadth due to the presence of one or more secondary (facultative) bacterial symbionts in their hosts, which hone co-evolution right down to the species genotype level. In aphids, the antifeedants and other chemicals and to some extent anatomical features (e.g. trichomes) hone co-evolution, certainly in tansy aphids (for which new data exists), down to the level of different parts of the plant, thereby creating different ecological habitats within the same individual plant! If all this is true, how can aphids or their wasp parasitoids be generalist? We have to find another term and at least for now, recommend using only levels of phagy.
The RES President, Chris Thomas FRS, briefly discussed this matter with one of us (H.D.L.) at the recent RES Ento ‘19 meeting held in London and said that he didn’t like using any specific ‘pigeon hole’ terms for defining diet breadth in insects. This is all very well, but the fact remains that people do use these terms and if we take generalism away from them, we have to give them something to use. Indeed, ecologists are always using terms to define categories of traits, such as ‘habitat or niche specialist’. There is no reason why the use of similar terminology cannot be applied to insects, or indeed any animal taxa. Since the doyen of aphidology, Roger Blackman, himself uses levels of phagy to describe what aphids consume in terms of dietary range, then we think this a safe choice….for now at least.
If others wish to countermand our interpretation of the existing published literature, then we are very happy that they do so. By this means, hopefully and ultimately, we may collectively get nearer to the truth of this matter regarding so-called generalism vs. specialism. For the time being, all we can do is make our case and stand back and see what happens, i.e. whether fellow entomologists, and biologists more widely, embrace our views, or not as the case may be. But if they do not, then they surely have to come up with an alternative framework wrought from the existing molten pool of available knowledge.

Reviewer 3 Report
The authors of this manuscript provide convincing arguments for restricting the use of terms describing diet/host breadth to “levels” of phagy, especially for aphids and parasitoids.
It would be hard to argue based on available evidence that the direction of diet breadth evolution is not primarily towards specialization, especially for aphids and their parasitoids. It could be argued, however, and probably should be in this paper that increases in diet breadth for some major insect taxa were positively correlated with speciation. The evolution of extra-oral digestion among certain groups provides at least a conceptual argument that increasingly polyphagous species fill new niches.
Possibly one aspect of the topic not developed has to do with the quantitative analyses implications of describing phagy, versus utilization of the categorical groups recommended by the authors.
The authors use the inevitable directionality associated with speciation and temporary filling of a niche as a fundamental argument to exclude the term of generalist that may be appropriate for a population at a particular space and time where food availability and utilization are not meaningful selective forces. The diet breadth at that time-space event of non-limiting food resources may be quite broad. They claim this “relaxed selection” likely only exists in unique situations such as in agroecosystems, but state that relaxed selection does not necessarily exist for Grizzly bears due to other evolutionary factors such as behavior, which in my opinion, does not strengthen their argument in this particular manuscript. Perhaps, the authors should discuss diet breadth evolution among Ursus species to provide a more compelling analogous argument. Perhaps the authors should also consider detailed examples from marine ecosystems, since the title is “Generalism in nature…”
Relative to the central argument, details associated with pest management factors were often difficult to interpret.
Ultimately, and based on arguments discussed in the manuscript, I think the authors have more than just a semantics argument, and publication of this concept should result in further critical review.
Author Response
Referee 3
As with Referee 1 and 3, we are grateful to the referee for reading our manuscript and giving us the benefit of their wisdom in term of its suitability of publication. We are naturally pleased that the referee is in agreement that this is an important topic for discussion and hence publication in the Cutting edge evolution of insects special issue of Insects and hence in the public forum
“Ultimately, and based on arguments discussed in the manuscript, I think the authors have more than just a semantics argument, and publication of this concept should result in further critical review.”
Reviewer: ‘It would be hard to argue based on available evidence that the direction of diet breadth evolution is not primarily towards specialization, especially for aphids and their parasitoids. It could be argued, however, and probably should be in this paper that increases in diet breadth for some major insect taxa were positively correlated with speciation. The evolution of extra-oral digestion among certain groups provides at least a conceptual argument that increasingly polyphagous species fill new niches.’
Authors: To be honest, when a species expands its dietary breath, then populations become isolated and specialized on new plant foods, reducing or even eliminating gene flow among populations feeding on different plants. So an increase in dietary breath creates specialism at both the population and eventually species level. We believe that we have shown that certainly in our two groups, aphid and wasp parasitoids, the central thrust is towards finer and finer grained levels of specialisation due to chemical-biochemical-physiological-immunological constraints, and that high levels of polyphagy are possible in the agroecosystem because we humans breed plants with reduced titres of antifeedant. Hence, the aphids, used as they are to normally dealing with high titres of antifeedant/s, co-evolved with over countless ages, have less of a challenge ecologically-speaking to increase their diet breath….but as Table 1 reveals, only slightly and certainly not as much as hitherto imagined, including by us.
Reviewer: ‘Possibly one aspect of the topic not developed has to do with the quantitative analyses implications of describing phagy, versus utilization of the categorical groups recommended by the authors.’
Author: We think this a good idea and maybe food for thought (so to speak) regarding the planned fourth paper in our series of papers on this fascinating topic.
Reviewer: ’The authors use the inevitable directionality associated with speciation and temporary filling of a niche as a fundamental argument to exclude the term of generalist that may be appropriate for a population at a particular space and time where food availability and utilization are not meaningful selective forces. The diet breadth at that time-space event of non-limiting food resources may be quite broad.’
Authors: We argue against this because of the many constraints placed on aphids and wasp parasitoids being able to veer from their ancient, co-evolved associations of aphid-plant and aphid parasitoid, respectively.
Reviewer: ‘They claim this “relaxed selection” likely only exists in unique situations such as in agroecosystems, but state that relaxed selection does not necessarily exist for Grizzly bears due to other evolutionary factors such as behaviour, which in my opinion, does not strengthen their argument in this particular manuscript.’
Authors: We argue that the agroecosystem is to some degree not the ‘real world’, as we humans breed plants with lower antifeedant titres, because we get higher yielding crops and we usually prefer such crops as they taste nice, which is what the aphids also seemingly prefer and hence face less of a challenge in terms of shifting host/s in such a situation, whilst perhaps also entering into enemy free space, away from their long co-evolved associations with antagonists. We acknowledge that aphids and wasp parasitoids do definitely use chemical cues and behaviour to locate their preferred host/s, often plant kairomones in the case of the latter.
Reviewer: ‘Perhaps, the authors should discuss diet breadth evolution among Ursus species to provide a more compelling analogous argument.’
Authors: We have decided to eliminate the Grizzly bears and other mammals from our discussion so as not to go too far off-piste from our original vision; after all, this is an ‘Insects’ paper. But surely Grizzly bears are very specialist animals indeed in terms of their anatomy, chemistry-biochemistry, physiology and behaviour. For example, what large animals except bears can hibernate for months on end and stand in freezing torrents of river water to catch salmon? They have specialised fur, teeth, claws and hair to do all the things they need to do and are anyway often ’serial specialists’, searching for different types of food at different times of the year, both in terms of availability and need (i.e. salmon towards the point where they need to think about hibernating). But a nice point for sure and one worthy of further discussion. Perhaps an international meeting needs to be convened to discuss this topic in greater detail using a range of different animal taxa.
Reviewer: ‘Perhaps the authors should also consider detailed examples from marine ecosystems, since the title is “Generalism in nature…” ‘
Authors: This we feel this is beyond the scope of our present article, which is again about insects for an entomological journal.
Reviewer: ‘Relative to the central argument, details associated with pest management factors were often difficult to interpret.’
Authors: We agree with this point and have tried to better focus our arguments.
